# Modification with Conventional Surfactants to Improve a Lipid-Based Ionic-Liquid-Associated Transcutaneous Anticancer Vaccine

**DOI:** 10.3390/molecules28072969

**Published:** 2023-03-27

**Authors:** Shihab Uddin, Md. Rafiqul Islam, Rahman Md. Moshikur, Rie Wakabayashi, Muhammad Moniruzzaman, Masahiro Goto

**Affiliations:** 1Department of Applied Chemistry, Graduate School of Engineering, Kyushu University, 744 Motooka, Nishi-ku, Fukuoka 819-0395, Japan; 2Department of Applied Chemistry and Chemical Engineering, Noakhali Science and Technology University, Noakhali 3814, Bangladesh; 3Advanced Transdermal Drug Delivery System Center, Kyushu University, 744 Motooka, Nishi-ku, Fukuoka 819-0395, Japan; 4Department of Chemical Engineering, Universiti Teknologi PETRONAS, Seri Iskandar 32610, Perak, Malaysia; 5Division of Biotechnology, Center for Future Chemistry, Kyushu University, 744 Motooka, Nishi-ku, Fukuoka 819-0395, Japan

**Keywords:** lipid-based ionic liquid, surfactants, transdermal drug delivery, nanovaccination, immunoresponse, tumor challenge, and biosafety

## Abstract

Transcutaneous vaccination is one of the successful, affordable, and patient-friendly advanced immunization approaches because of the presence of multiple immune-responsive cell types in the skin. However, in the absence of a preferable facilitator, the skin’s outer layer is a strong impediment to delivering biologically active foreign particles. Lipid-based biocompatible ionic-liquid-mediated nanodrug carriers represent an expedient and distinct strategy to permit transdermal drug delivery; with acceptable surfactants, the performance of drug formulations might be further enhanced. For this purpose, we formulated a lipid-based nanovaccine using a conventional (cationic/anionic/nonionic) surfactant loaded with an antigenic protein and immunomodulator in its core to promote drug delivery by penetrating the skin and boosting drug delivery and immunogenic cell activity. In a follow-up investigation, a freeze–dry emulsification process was used to prepare the nanovaccine, and its transdermal delivery, pharmacokinetic parameters, and ability to activate autoimmune cells in the tumor microenvironment were studied in a tumor-budding C57BL/6N mouse model. These analyses were performed using ELISA, nuclei and HE staining, flow cytometry, and other biological techniques. The immunomodulator-containing nanovaccine significantly (*p* < 0.001) increased transdermal drug delivery and anticancer immune responses (IgG, IgG1, IgG2, CD8+, CD207+, and CD103+ expression) without causing cellular or biological toxicity. Using a nanovaccination approach, it is possible to create a more targeted and efficient delivery system for cancer antigens, thereby stimulating a stronger immune response compared with conventional aqueous formulations. This might lead to more effective therapeutic and preventative outcomes for patients with cancer.

## 1. Introduction

Vaccination is the most successful immunotherapeutic approach in modern medical sciences, as it prevents 3.5–5 million deaths annually, whereas 1.5 million each year people die of diseases that can be prevented by immunization (WHO-2022) [1,2,3]. Transcutaneous vaccination is a patient-friendly, effective, and target-specific alternative vaccination/immunization approach because skin contains numerous immunoresponsive antigen-presenting cells (APCs), which are highly efficacious for immunoresponsive diseases such as cancer, anemia, HIV, and diabetes [4,5]. However, the lipidic arrangement of the stratum corneum (SC) hinders the permeation of large and hydrophilic molecules (proteins, peptides, and nucleic acids), which must be transported using a hydrophobic, lipophilic, or oil-based nanocarrier [6,7,8,9]. Ionic liquids (ILs), which are liquid salts with a melting temperature below 100 °C, can be used as reagents, solvents, and antisolvents in the synthesis and crystallization of active pharmaceutical ingredients (APIs); as solvents, cosolvents, and emulsifiers in drug formulations; as pharmaceuticals (API-ILs) aiming at liquid therapeutics; and in the development and/or improvement of drug-delivery-based systems [10]. Recently, new developed lipid-based biocompatible ionic liquids (LBILs), actively employed in nanodrug formulations (ILNDFs), represent an emerging modality for overcoming these challenges and improving transdermal drug delivery systems (TDDSs) based on the lipophilic and wide solubility of LBILs [11,12]. Multiple conventional surfactants have been used with ionic liquids (ILs) to increase the effectiveness of IL-based formulations for transdermal drug delivery because these surfactants act as emulsifiers, wetting agents, dispersants, and foaming agents that lower surface tension and diffuse the lipidic arrangement of the SC to improve drug delivery, pharmacokinetics, and pharmacodynamics [11,13,14,15]. However, to generate an effective IL-based formulation, the most appropriate combination of ILs and surfactants is essential [13,16]. Moreover, to develop transcutaneous vaccines, it is also necessary to activate the adaptive immune system (AIS), which is the most crucial step in developing a transcutaneous vaccine triggered by antigen uptake by skin dendritic cells (DCs) with appropriate kinetics [17,18]. There are only three pharmaceutically approved immunomodulators (imiquimod (IMQ), bacillus Calmette–Guerin, and monophosphoryl lipid-A), and these antigens target Toll-like receptors to stimulate the innate immune response in the human body [19,20]. These immunomodulators, which can activate APCs in the skin, are employed as adjuvants with antigens or antigenic proteins in vaccine formulations to enhance the immune response [19,20,21]. Among them, IMQ is an excellent immunostimulatory adjuvant that has been used in various immunotherapies, but IMQ-based formulations are challenging to use for transdermal drug delivery because of their limited solubility and low permeability [4,22,23,24,25]. LBIL-associated formulations might represent a promising TDDS approach to overcome these challenges through the generation of nanocarriers in isopropyl myristate (IPM) in the oil phase. In our previous studies, we found that LBILs significantly increased protein and peptide delivery, whereas surfactants improved formulation quality and drug delivery [4,26]. However, the tests revealed that the immune response to LBIL-based formulations was inferior to that of injectable aqueous formulations. Additionally, the impact of different surfactants in LBIL-based formulations was not investigated in earlier studies [4,12]. Because of the limitations of our previous studies, we designed an LBIL- and conventional surfactant-based nanovaccine formulation containing an immunomodulator (IMQ) and the high-molecular-weight protein ovalbumin (OVA) in IPM [4].

In this paper, for the first time, we report the effects of various conventional surfactants (cationic, anionic, and nonionic) in IL-based formulations in IPM, their cellular toxicity, and drug release profile. We discovered that nonionic surfactants improved formulation effectiveness and optimized transdermal drug delivery. We used a freeze-drying emulsification approach to produce the most effective nanovaccine containing a surfactant (Span-20 (S-20)), an LBIL-associated antigenic protein (OVA), and an immunomodulator (IMQ) in IPM. Transdermal patches were also tested for drug delivery and therapeutic and preventive effect, and anticancer immune responses in OVA-specific tumors generated by inoculating EG7-OVA cells into C57BL/6N mice. Compared with an aqueous formulation, the transcutaneous nanovaccine significantly (*p* < 0.001) increased transdermal drug delivery, pharmacokinetics, therapeutic effects, and prophylactic effects in C57BL/6N mice without causing biological toxicity. Furthermore, the transcutaneous nanovaccine strongly (*p* < 0.001) suppressed tumor growth and development by significantly (*p* < 0.0001) stimulating anticancer immune responses, and it outperformed a subcutaneously injected aqueous nanovaccine in terms of the immune response. The nanovaccine boosted the expression of anticancer antibodies such as immunoglobulin G (IgG, including IgG1 and IgG2a) in blood plasma, the levels of dermal DC (dDC)-presenting keratinocyte-deriving antigens (CD207+, CD103+), and the counts of cancer-suppressive cytokine cytotoxic T-cells (CD8+ T-cells) in the tumor microenvironment (TME). Enzyme-linked immunosorbent assay (ELISA), 4,6-diamidino-2-phenylindole (DAPI) staining, and multichannel flow cytometry were used in these investigations. Finally, the skin irritation test and histopathological investigations including the 3-(4, 5-dimethylthiazol-2-yl)-2, 5-diphenyltetrazolium bromide (MTT) assay and hematoxylin–eosin (H&E) staining validated the biological safety of the nanovaccine in human artificial 3D-LabCyte EPIMODEL cells and C57BL/6N mice. Considering these findings, we speculate that the nanovaccine containing an LBIL-associated protein and immunomodulator is a transcutaneous anticancer vaccine worthy of further study.

## 2. Results and Discussion

### 2.1. Selection of Surfactants for IL-Based Formulations

The ingredients of pharmaceutical drug formulations play a significant role in improving formulation stability, drug solubility and encapsulation, and drug permeation through the skin. The selected constituents of a formulation should be nontoxic and biocompatible for effective transdermal drug delivery [27]. In our previous study, LBILs were found to be completely miscible with IPM (considered a safe and pharmaceutically accepted ingredient in drug formulations), and they exhibited excellent biocompatibility compared with conventional surfactants [28]. An IL/O-ND composed of IL[EDMPC][Lin]/S-20 at a 5:5 wt.% ratio displayed maximum peptide encapsulation with excellent stability [25], whereas a 2.5:2.5 wt.% ratio of IL[EDMPC][Lin]/T-20 provided the highest transdermal permeation of the antigenic protein OVA [4]. An equal ratio of IL and cosurfactant resulted in better transdermal drug delivery output, and thus, we also selected a series of cosurfactants, including cationic, anionic, and nonionic surfactants, to prepare IL[EDMPC][Lin]/cosurfactant formulations (ILNDFs) with OVA at a weight percent ratio of 2.5:2.5 to investigate the effect of cosurfactants in drug formulations as well as TDDSs. Additionally, we designed a nanovaccine system that coupled the FDA-approved immunomodulator IMQ with ILNDFs to boost the prophylactic/preventive, therapeutic, and immune response capabilities of the vaccine. LBILs play a vital role in developing acceptable IL-based formulations, whereas conventional surfactants strengthen the stability of the formulations [4,11]; therefore, ILs were examined as surfactants.

### 2.2. Morphological Behaviors and Cytotoxicity of ILNDFs

The hydrophobic nature of skin prevents the passage of hydrophilic antigens with molecular weights exceeding 500 Da through an aqueous system. Conversely, hydrophilic molecules are not dissolved or dispersible in such a hydrophobic phase [29]. Hydrophilic OVA was disseminated and stabilized in a hydrophobic IPM solution in ILNDFs utilizing LBIL-([EDMPC][Lin]) and cationic/anionic/non-ionic conventional surfactants at a 1:1 w/w ratio (Figure 1A). DLS was used to measure the particle size (n/z) and PDI of the ILNDFs (summarized in Table 1), and the particle size was approximately or below 500 nm in all formulations (Appendix A), which is suitable for TDDSs [4,29,30]. The CLSM observation confirmed that the FITC-OVA nanoparticles moved freely in the IPM, and the particle size agreed well with the DLS observation (Appendix A). Because of the ionic properties of cationic and anionic cosurfactant-based formulations, interfacial-tension-reducing capability is significantly affected, leading to physical instability at room temperature (25 ± 1 °C) [31],and precipitation within 1 week, whereas nonionic cosurfactant-based formulations, apart from Brij-35- and PEG-based formulations, were stable (Appendix A) [31]. The hydrophobic and hydrophilic structural groups of nonionic surfactants aid in correctly dispersing drug molecules in solution as emulsifiers or foaming agents, whereas ionic surfactants work in the reverse way, which leads formulation instability [32]. Nonionic cosurfactants improved the phase stability with the IL and drug by forming covalent or weak van der Waals interactions in the oil phase, resulting in increased bioactivity and physiochemical stability for no-ionic ILNDFs [31]. Ionic surfactants can transfer their cationic or anionic charge to the surface of the nanoparticles in liquid formulation, which leads the nanoparticles to deteriorate and precipitate due to electrostatic forces; as a result, ionic surfactant-based formulations become unstable [33]. Nonionic surfactants can lessen harshness, making them skin-friendly. On the other hand, Ionic surfactants can be harsh and irritating to the skin due to their interaction with the predominantly negatively charged ions of the skin. [32]. Only nonionic cosurfactant-based formulations were able to sustain particle size stability for 3 months (Appendix A). The ionic properties of both cationic and anionic surfactants triggered precipitation, which led to instability. The interface attraction of surfactant and cosurfactant molecules acts as a steric or electrostatic barrier to droplet particle agglomeration, consequently boosting formulation stability [34].

The hydrophobic tails and hydrophilic heads of nonionic surfactants allow drug molecules to dissolve in the oil phase, thereby improving stability and biocompatibility. Furthermore, cationic and anionic surfactants carry positive and negative charges in a medium, respectively, causing damage to the cell membrane or a toxic effect and resulting in reduced cell viability. The safety of a new surfactant must be demonstrated in nonclinical and clinical studies according to U.S. FDA guidelines, especially when dealing with pharmaceutical formulations [35]. No-ionic ILNDFs were associated with greater cell viability in skin irritation tests using the LabCyte EPI model cell line, whereas both cationic and anionic ILNDFs had toxic effects. Cell viability was significantly reduced (*p* < 0.01) when PBS solutions were compared with BKC-, DA-, SS-, Brij-35–, DTAB-, SDS-, and MBS-based ILNDFs; however, S-20-, S-80-, T-20-, T-80-, squalene-, and PEG-based ILNDFs were biocompatible (Figure 2C).

### 2.3. Effect of ILNDFs on Drug Encapsulation and Skin Permeation

Nonionic surfactants have some advantages over ionic surfactants in that they can achieve a wide range of hydrophilic–lipophilic balance (HLB) by changing their molecular structure, in addition to their hydrophilic moiety, high degree of size customization, and lower critical micelle concentration, which are favorable for emulsification in the oil phase [36]. The HLB value represents the surfactant’s propensity to incorporate in water or oil and form an emulsion. Surfactants with low HLB values tend to be more soluble in oil, whereas those with high HLB numbers tend to be more soluble in water [37,38]. Those properties of nonionic surfactants allowed them to load higher amounts of drug than ionic surfactants (Appendix A). Compared with surfactant-free formulations, all nonionic surfactants showed significantly increased (*p* < 0.001) drug-loading capacity, whereas ionic surfactants had no such effect. Nonionic surfactants have varying HLB values (e.g., S-20 = 8.6, S-80 = 4.3, T-20 = 16.72, T-80 = 15, Brij-35 = 16.9, squalene = 8.4, PEG = 20), and a reasonable range for emulsification is between 3.6 and 8.0 [33,37,38]. Due to S-20’s HLB being in a good range (HLB = 8.6) for emulsification formulations, which enabled S-20 to develop more sustainable formulations, as well as the possibility that S-20 has a potentially better drug-loading capacity compared to Tween and other surfactants. (Figure 2A) [38]. However, it is currently challenging to adequately describe how LBILs may affect pharmaceutical formulations, because the HBL values of LBILs have not yet been studied. To fully understand the mechanistic effect of LBILs in oil-based drug delivery systems, more research is required. Surfactants have the ability to alter the permeability of cellular or biological membranes by concentrating at phase interfaces, resulting in lower membranous interfacial tension with increasing duration of skin contact, which influences transdermal drug delivery [34,36]. Both cationic and anionic ILNDFs increased transdermal drug delivery (Appendix A), albeit without significance (Appendix A), whereas nonionic ILNDFs, excluding Brij-35-based formulations, dramatically boosted transdermal and topical drug delivery (Figure 2B, Appendix A). All nonionic ILNDFs significantly improved controlled transdermal drug delivery into the skin (*p* < 0.01) compared with aqueous formulations in the order of S-20 > S-80 > PEG > T-20 > T-80 > squalene > Brij-35 (Figure 2 and Appendix A). The head group of the cosurfactant had a significant impact on overcoming the cutaneous barrier, and S-20 was more hydrophobic than T-20, allowing it to more effectively increase 5-fluorouracil penetration (LogP: 4.26 vs. 3.72) [39]. Based on the in vitro transdermal drug delivery results, the skin permeation parameters were calculated using lag time methods, and it was discovered that the transdermal flux was 20-fold higher in the S-20-based formulation. Compared with the aqueous formulation, the permeation coefficient, diffusion coefficient, and skin partition coefficient were significantly higher for S-20-based ILNDFs, which displayed the best performance among all ILNDFs (Table 2). Based on skin permeation, formulation stability, toxicity, and drug-loading capability, we avoided ionic ILNDFs and continued our nanovaccine study with the best combination: LBIL and S-20.

### 2.4. Morphological and Transdermal Delivery of the Nanovaccine

Using DLS, the nanoparticle size of the nanovaccine in IPM solution (Figure 1B) was examined, and one identical peak was identified (Appendix A). The encapsulation of IMQ and OVA by LBIL within the core of the nanoparticle was confirmed by DLS because the particle size of the immunomodulator-containing formulations (IL/NP (+)) was larger than that of non-IMQ-based formulations (IL/NP (−)) (Appendix A). The antigenic protein (OVA) and immunomodulator (IMQ) were both physiochemically stable in the nanovaccine system for up to 3 months at room temperature (25 ± 1 °C), as measured by the particle size and the FITC-OVA and IMQ concentrations (Appendix A).

OVA and IMQ were encapsulated with LBIL and S-20 in the nanovaccine system (IL/NP), which accelerated transdermal penetration and formulation stability. Only the IL-surfactant (LBIL)-based formulation IL/NP(+) @IMQ had less effective transdermal drug delivery and formulation stability than the formulation coated with surfactant (LBIL) and cosurfactant (S-20), IL/NP(+) (Figure 3A(i,ii) and Figure 3B(i)). The findings demonstrated that the S-20 cosurfactant in the formulation enhanced the stability of the drug formulation as well as the drug release profile. Compared with the findings for the aqueous formulation, IL/NP significantly (*p* < 0.001) boosted in vivo transdermal and topical delivery as well as the depth of drug penetration into YMP skin (Figure 3A). Additionally, we used a homemade transcutaneous patch to confirm the in vivo drug permeation capacity of the nanovaccine in C57BL/6N mice. By measuring the topical delivery of OVA and IMQ and the depth of FITC-OVA penetration into skin, we discovered that the nanovaccine enhanced the delivery of both FITC-OVA and IMQ into the mouse skin (Figure 3B and Table 2). Transdermal drug delivery is highly facilitated by lipophilic and hydrophobic surfactants because they significantly alter the lipid bilayer structure of the skin without damaging the epidermis [40]. Using FTIR spectrometry, the impact of the nanovaccine on the SC was examined (Appendix A). IL/NP(−) displaced the intercellular lipid composition with gauche-trans conformational transitions in the largest symmetrical and asymmetrical order, which improved drug permeation through the intercellular lipidic space (Figure 3C). There are three possible routes of transdermal drug delivery: intracellular, transcellular, and intercellular [22,39]. Furthermore, the CLSM of the skin on mouse ears revealed that FITC-OVA was present on the corneocyte edge and indicated absorption via the intercellular route, which was strongly associated with the nanovaccine’s ability to alter the lipidic barrier of the SC (Figure 3D).

### 2.5. Prophylactic Effect of the Nanovaccine against Tumors

We vaccinated mice with three successive doses given 7 days apart using transdermal patches and subcutaneous injection to assess the prophylactic effect of the nanovaccine and its activation of the adaptive immune response. Before administering OVA-specific tumor growth cells to prevaccinated mice, the antitumor immune response was assessed with ELISA, and the animals were considered dead when the tumor volume reached 2500 mm^3^ (Figure 4A). We also examined tumor growth and development rates, body weight variations, and mouse survival rates. Whereas the plasma drug level and subsequent immune response fluctuate with traditional delivery systems, thereby preventing a sustained therapeutic response, controlled delivery systems facilitate target-specific delivery and precisely achieve maximal efficacy and immune responses [41]. Using a subcutaneous injection technique, drug concentrations in the blood were temporarily enhanced, whereas LBIL-based formulations delivered via a transcutaneous patch facilitated controlled drug delivery and achieved better pharmacokinetic and pharmacodynamic properties than injections [11,25]. In addition to triggering the immune system, immunomodulators that incorporate the IL/NP system logically promote the innate immune response in response to tumor challenge. The immune response was improved by immunomodulator-containing compared with the effects of immunomodulator-free formulations and aqueous formulations, and the innate responses of IgG, IgG1, and IgG2a decreased in the order of IgG (+) transdermal patch > IgG (+) subcutaneous injection > IgG (−) transdermal patch > aqueous solution (Figure 4B). Compared with the aqueous formulation, IL/NP(+) transdermal patches significantly suppressed tumor growth and development and body weight variation and increased mouse survival, and the trends were similar to those of the antigenic IgG immune response (Figure 4C). These intriguing results support the efficacy of the skin-targeting immunization strategy, termed “transcutaneous vaccination” [14].

### 2.6. Therapeutic Immunization and Tumor-Suppressive Effects of the Nanovaccine

By conducting additional studies on the therapeutic impact in C657BL/6N mice using the same strategy, the prophylactic efficacy of the nanovaccine was validated. Prior to antitumor studies, mice carrying tumors induced by EG7-OVA cells were transdermally and subcutaneously immunized three times (Figure 5A). In comparison with the aqueous formulation, transcutaneous immunization significantly reduced tumor growth and development (Figure 5B), prevented body weight variation (Figure 5C), and boosted mouse survival (Figure 5D). Meanwhile, transdermal patches outperformed subcutaneous injection in terms of cancer prevention. In this study, we looked at how the nanovaccine affected tumor growth and development, but we did not look at whether it was sufficiently successful or able to elicit a strong enough immune response to completely eradicate the tumor, as is typical for an anticancer vaccine. We explored the immunological (IgG) response in C57BL/6N mice generated by vaccination because IgG plays a vital role in various types of malignancies. We demonstrated that IL/NP(+) significantly increased plasma IgG expression, which was consistent with the prophylactic effect of nanovaccination (Figure 5E–G). Tumor-driven IgG is related to the growth and survival of cancer cells, and it facilitates the metastatic process and helps to developed anticancer antibodies [42]. IgG1 and IgG2a expression is beneficial, and it is directly related to the T-helper cell type (Th1 and Th2) autoimmune responses, which are associated with antitumor responses [21,43,44]. The topically applied IL/NP(+) formulation induced the highest levels of total IgG, IgG1, and IgG2a expression, outperforming subcutaneous IL/NP(+) and transdermal IL/NP(−) in terms of the immune response. These noteworthy findings indicated that a transcutaneous anticancer vaccine was successfully developed by combining an immunomodulator and antigenic protein [45]. However, more research is necessary to make it a more effective anticancer medicine for clinical trials.

### 2.7. Antitumor Immune Response Induced by the Nanovaccine

We used flow cytometry to assess the quantitative antigen uptake by skin DCs to validate the prophylactic and therapeutic efficacy of the nanovaccination. Activation of the AIS is the most crucial step in developing a successful transcutaneous vaccine, and AIS activation is triggered by antigen uptake by skin DCs [14]. We emphasized the CD207+ (which includes epidermal LCs and dermal DCs) and CD103+ (which include dermal DCs rather than epidermal LCs) subsets because they have the ability to present antigens expressed by skin keratinocytes in an antigen cross-presentation manner. These subsets are closely related to Th1-type adaptive immunity and are consequently linked to anticancer activity [42,43]. Antigen delivery to LCs and CD103+ dDCs was investigated and calculated using the antigen uptake ratio of these skin subsets. Skin dDCs and LCs were identified as the CD207+ cell populations and marked as (+/+) and (+/−), respectively, on the cytograms, where the vertical and horizontal axes represent the fluorescence intensities of Cy5-OVA and Cy7-anti-CD103 antibodies, respectively (Figure 6A). For CD207+ cells, compared with the aqueous formulation, antigen uptake was significantly increased (*p* < 0.0001) by IL/NP(+) transcutaneous vaccination, with the effect being stronger than that induced by the injectable IL/NP(+) and transdermal IL/NP(−) formulations, which signified the lengthening of the dendritic response to the nanovaccine as a result of physiochemical stimulation (Figure 6B). Furthermore, because of the significant penetration-enhancing effects of IL/NP (+), CD103+ dDCs, also known as CD207+ cells, had a significantly higher uptake ratio (Figure 6C). The fluorescence peak intensity of APCs in the IL/NP groups was higher than that in the control group, demonstrating the effectiveness of the nanovaccine as an antigen carrier that promotes antigen uptake by DCs (Figure 6D).

The backbone for current successful cancer immunotherapies is the activation of cytotoxic T-cells (CD8+ T-cells), which are the most efficient immune effectors of anticancer activity [43]. In tumors, cytotoxic CD8+ T cells produce perforin and granzymes, which can kill cancer cells by releasing cytokines such as IFN and TNF. IFN can inhibit cellular replication and stimulate cancer cell macrophages, and it can synergize with TNF-α in macrophage activation, allowing CD8+ T cells to prevent cancer [46]. To investigate the activation of CD8+ T cells in the TME, we induced tumor formation in C57BL/6N mice by injecting them with EG7-OVA cells. After tumor budding, the mice were vaccinated with three consecutive doses, and 7 days following the third dose, the tumor was harvested for bisectioning and cryosectioning. To visualize CD8+ T cells in the TME, sectioned tumor tissues were stained with DAPI (Figure 7A). From the microscopic images of stained tissues, the number of CD8+ T cells was calculated using ImageJ software. The number of CD8+ T cells was remarkably higher following transcutaneous vaccination (Figure 7B,C) than after vaccination with the aqueous formulation, with the effect increasing in the order of control < transdermal IL/NP(−) < subcutaneous IL/NP(+) < transdermal IL/NP (+). This finding is in good agreement with the prophylactic, therapeutic, and immune-activating effects of the nanovaccine.

### 2.8. Biological Safety of the Nanovaccine

Biocompatibility is an essential requirement for a novel pharmaceutical formulation (vaccine) because it has a significant impact on quality of life [47]. Biocompatible formulations must be harmless to both cell populations and bodily organs (such as the skin, liver, kidneys, spleen, heart, and lungs) that are directly contacted by the vaccine or are associated with drug distribution and metabolism [48,49]. Aiming to develop a biocompatible transcutaneous vaccine, we evaluated the toxicological impact of the vaccine on the human artificial 3D-LabCyte EPIMODEL cell line, which is the most well-known cell line for examining candidate TDDSs. We found that our newly developed nanovaccine was completely nontoxic to the cell line and as biocompatible as PBS (Figure 8B).

Furthermore, we investigated the biological safety of the nanovaccine in various organelles associated with TDDS in C57BL/6N mice. Following the administration of three successive vaccine doses, we harvested organs (the skin, liver, kidneys, spleen, and lungs) for histopathological analysis using H&E staining (Figure 8A) [50]. During this experimental period, we found that the nanovaccine had no adverse effects on mouse survival rates or body weight variation (Figure 8C). The histograms of the H&E staining of the skin, liver, kidney, heart, spleen, and lung tissues in normal mice (control group) and vaccinated mice (IL/NP(+) and IL/NP(−) groups) were similar, indicating no negative effects on tissues excluding skin SDS-based vaccine-treated mice (Figure 8D). Although SDS has known hazardous effects on the epidermis, SDS-based vaccination had no adverse effect on drug-metabolizing organs tissues, indicating the biosafety of the antigen and immunomodulator in the body. No toxic effects were observed in the skin and organ histopathology study in all nanovaccine treatment groups, further proving the biological safety of the nanovaccine.

## 3. Experimental Materials and Methods

### 3.1. Chemicals and Reagents

The OVA from chicken egg white, linoleic acid, IPM, bis (3-aminopropyl) dodecylamine, S-20, Span-80 (S-80), Tween-20 (T-20), Tween-80 (T-80), ethylenediaminetetraacetic acid (EDTA), 50 U/mL DNase I, and mounting medium for microscopic entellus were purchased from Sigma-Aldrich (St. Louis, MO, USA). IMQ, 1,2-dimyristoyl-sn-glycero-3-phosphocholine (DMPC), ethyl trifluoromethanesulfonate, 1-dodecyl-3-methylimidazolium bis(trifluoromethylsulfonyl)imide, isopropyl alcohol, diethylene glycol mono-ethyl ether (as a chemical permeation enhancer), 0.4% paraformaldehyde, chloroform super-dehydrate, benzalkonium chloride (BKC), dodecyltrimethylammonium bromide (DTAB), sodium dodecyl sulfate (SDS), squalene, and methyl benzenesulfonate (MBS) were purchased from Tokyo Chemical Industry Co. Ltd. (Tokyo, Japan). Hematoxylin, eosin, acetic acid, methanol, ethanol, acetonitrile, sodium stearate (SS), bis(3-aminopropyl) dodecylamine (DA), and polyethylene glycol (PEG) were purchased from Fujifilm Wako Pure Chemicals Industries Ltd. (Osaka, Japan). Fluorescein isothiocyanate (FITC) was purchased from Thermo Fisher Scientific (Waltham, MA, USA). Brij-35 was purchased from Kishida Chemical Co. Ltd. (Osaka, Japan). All other chemical reagents used in these studies were of analytical grade.

Frozen mouse skin and YMP skin were purchased from Hoshino Laboratory Animals (Ibaraki, Japan) and stored at −80 °C. Female C57BL/6N mice (6 weeks old, weighing 20 ± 2 g) were purchased from Kyudo (Saga, Japan) and housed in a controlled environment. The human artificial 3D epidermis cell line LabCyte EPI-Model and MTT assay medium were purchased from Japan Tissues Engineering Co. Ltd. (Aichi, Japan). Research-grade mouse antiserum CD8+ monoclonal antibody, mouse antiserum CD4+ monoclonal antibody, CD207 antibody, fetal bovine serum, Roswell Park Memorial Institute 1640 Medium, and all other related reagents, as per the protocols, were purchased from Thermo Fisher Scientific. Cy5-monoreactive dye pack was purchased from GE Healthcare (Chicago, IL, USA). PE/Cy7-anti-CD103 antibody was purchased from BioLegend (San Diego, CA, USA). Liberase was obtained from Roche (Basel, Switzerland). Mouse DC2.4 dendritic cells and mouse E.G7-OVA OVA-specific melanoma cells were purchased from the RIKEN Cell Bank (Tsukuba, Japan).

### 3.2. Preparation and Morphological Characterizations of ILNDFs

OVA antigenic protein was labeled with FITC and dissolved in Milli-Q water. Then, 25 mg of LBIL in cyclohexane were homogenized in 1 mL of OVA aqueous solution (containing 2.0 mg of OVA) and 25 mg of various cosurfactants (Table 1) with a mechanical homogenizer (Polytron; PT2500E). Drug–IL–cosurfactant complexes were prepared by freeze-dry emulsification and dispersed in 1 mL of IPM with magnetic stirring for 12 h (Figure 1A). These complexes were named conventional surfactant- and IL-associated nanodrug formulations (IL-NDFs). The particle size and polydispersity index (PDI) of ILNDFs were investigated using 1 cm long quartz cells at 25 ± 1 °C with a 173° angle of a dynamic light scattering (DLS) system (Zetasizer Nano series ZS, Malvern WR141XZ, Worcestershire, UK). Confocal laser scanning microscope (CLSM) (LSM700; Carl Zeiss, Germany) images of FITC-OVA nanoparticles dispersed in IPM at 63× was used to determine the morphological structure of freely movable particles in IPM. The physiochemical stability of the particles was confirmed by visual observation and by measuring the particle size of ILNDFs by DLS at room temperature for 90 days and the concentration of FITC-OVA using a multiplate reader at a wavelength of 485–535 nm.

### 3.3. Drug Loading, Encapsulation, and Skin Permeation of ILNDFs

We developed ILNDFs containing an excess amount of OVA (10 mg/mL) and then centrifuged them at 8000× *g* for 5 min. The upper transparent solutions contained encapsulated FITC-OVA in the oil phase, which was then dispersed in the diluent (mixture of PBS:methanol:acetonitrile = 2:1:1 *v*/*v*) and again centrifuged to obtain free FITC-OVA in the diluent. A microplate reader was used to measure the absorbance of FITC-OVA using the slope–intercept equation.

The in vitro transdermal delivery of ILNDFs through YMP skin was demonstrated using the Franz diffusion cell system (FDCS). In brief, the FDCS receiver chamber was filled with 5 mL of phosphate-buffered saline (PBS) and maintained at 32.5 °C. The FDCS donor chamber was adjusted with pigskin (2 × 2 cm) and filled with 200 µL of the test solutions. To assess drug penetration through the skin, 100 µL of the medium was collected from the receiver chamber at 12, 24, 36, and 48 h, and an equal amount of medium was added at each time point for replenishment. The concentration of FITC-OVA was determined using a multiple microplate reader at a wavelength of 485–535 nm.

To accomplish topical delivery, the target skin area was washed with 20% ethanol and cut into 16 pieces before being extracted for 12 h at room temperature in 1 mL of diluent, and the absorbance of FITC-OVA was measured. Transdermal drug permeation parameters were calculated by the lag time of least-square formulations using the following equations:*D_S_* = l*s*^2^/6*t_L_* and *Ks* = (*Jl*)/(*Ds* × *Cd*).(1)
where *Ds* is the skin permeation diffusion coefficient, *ls* is the thickness of YMP, *t_L_* is the lag time, *J* is the transdermal flux, *Cd* is the concentration of the drug in the tested formulation, *D_SC_* is the diffusion coefficient, and *l* is the intercept of the *x*-axis and slope of the approximate line.

### 3.4. In Vitro and In Vivo Toxicity Assessments

The in vitro cellular toxicity of ILNDFs was investigated by a skin irritation test using the LabCyte EPIMODEL cell line and the MTT assay as described previously, with slight modifications [22]. The protocol is briefly described in the Appendix A.

To assess the in vivo toxicological effect, we administered three successive doses of the nanovaccine to C57BL/6N mice at 7-day intervals, and after the third dose, various organs were harvested for histopathological examination using H&E staining [20,21] (briefly described in the Appendix A). During this period, we observed the effects of the nanovaccine on body weight, survival, and various drug-metabolizing organs (skin, liver, kidneys, heart, spleen, and lungs) using H&E staining.

### 3.5. Preparation and Morphological Observation of the Nanovaccine

The nanovaccine was prepared in the same manner as the ILNDFs. In this study, a complex of FITC-OVA (an antigenic protein), IMQ (an immunomodulator), DMPC- and linoleic acid-based IL (EDMPC), and S-20 were blended by freeze-dry emulsification at a ratio of 2:1:25:25 *w*/*w* and dispersed in IPM, and the resulting solution was named “nanovaccine” (Figure 1B). The particle size, PDI, and formulation stability were confirmed with the DLS system as described in Section 3.2. The IMQ concentration was measured using a high-performance liquid chromatography (HPLC) system, as described in Appendix A.

### 3.6. TDDS of Nanovaccine

The in vitro transdermal delivery of the nanovaccine was examined using the FDCS as described in Section 3.3. The concentration of FITC-OVA was quantified using a multiplate reader, and the concentration of IMQ was quantified by HPLC with slight modifications (briefly described in the Appendix A). Subsequently, the depth of FITC-OVA delivery across YMP skin was investigated using skin cryosections, which were then visualized with CLSM and quantified with ImageJ software, bundled with 64-bit Java-8, Version 1.53t, as summarized in Appendix A.

To check the in vivo transdermal delivery, 100 µL of the tested sample (containing 200 µg of OVA) was applied to the cleaned dorsal skin area using laboratory-produced transdermal patches (Cathereep FS dressing tape, Tokyo, Japan, 1 × 1 cm) for 6 h. After treatment, the patches were removed, and the skin was washed with 20% ethanolic solution to remove nonpenetrated drug formulations from the skin surface, and then the desired skin (epidermis and dermis) area on which the drug was applied (1 × 1 cm^2^) was harvested. The targeted skin was cut into 16 pieces, and OVA was extracted in 1 mL of diluent with vigorous shaking for 12 h and then centrifuged at 8000× *g* for 10 min. The FITC-OVA concentration in the upper transparent solution was determined using a multiplate reader, and the IMQ concentration was determined via HPLC. In addition, the depth of FITC-OVA penetration into the skin was investigated by CLSM by skin cryosections as described in the Appendix A.

### 3.7. Drug Release Pathway and Effect on the Skin

#### 3.7.1. Drug Release Pathway

Ears were harvested from C57BL/6N mice, and 10 µL of the tested samples (nanovaccine formulations) was placed dropwise on the rear (inner) side and allowed to sit for 10 min. The ears were then rinsed with 20% ethanol until the remaining solutions were eliminated and placed on a slide glass to observe the routes of drug permeation by CLSM.

#### 3.7.2. Effect on the SC Layer

The epidermal layer was removed after heating YMP skin for 2 min at 60 °C and soaked in EDTA solution for 2 h before being washed with ethanol and water and dried. The epidermis was immersed in the tested samples for 20 min, washed with ethanol, and dried. The impact of the tested materials on the internal structure of the SC layer was investigated using Fourier transform infrared (FTIR) spectroscopy (Perkin Elmer, FTIR Spectrum Two; Tokyo, Japan) at a wavelength of 4000–400 cm^−1^.

### 3.8. Vaccination and Immune Response against the Tumor

To evaluate the effect of the nanovaccine in C57BL/6N mice, mice were immunized with three consecutive doses of 100 µL (containing 200 µg of OVA and 100 µg of IMQ) of the nanovaccine in 7-day intervals using transdermal patches and subcutaneous injection. Seven days after the third dose, blood was drawn to evaluate IgG-specific antibody titers (IgG, IgG1, and IgG2a) in the blood plasma using ELISA as described previously with slight modifications [14] (briefly described in the Appendix A). In vivo experiments were performed with the approval of the ethics committee for animal experiments of Kyushu University (Approval code: A19-349-0) and conducted according to the guidelines of the Science Council of Japan.

### 3.9. Prophylactic and Therapeutic Effects of the Nanovaccine against Tumors

To examine the prophylactic effect of the nanovaccine against tumors, mice were vaccinated with three consecutive doses as described in Section 3.8, and 7 days after the third dose, the mice were inoculated with 100 μL of OVA-specific tumor cells in an EG7-OVA suspension (2.0 × 10^6^ cells) to permit tumor development in the immunized mice. The effect of the nanovaccine was confirmed by measuring tumor growth and development, body weight variation, and survival rates, as briefly described in the Appendix A.

To determine the therapeutic effect of the nanovaccine, a tumor was developed by inoculating EG7-OVA cells in C57BL/6N mice. Within 7–10 days, tumor budding was noticeable, and then the mice were vaccinated with three consecutive doses of 100 μL of the tested sample using a transdermal patch and subcutaneous injection. Tumor growth and development were calculated by measuring tumor the volume using the following equation [14]:Tumor volume [mm^3^] = (length [mm]) × (width [mm])^2^ × 0.5.(2)

The antitumor effect was verified by investigating the body weight variations and survival rates of the mice, as briefly described in the Appendix A. Furthermore, the antitumor immune response was evaluated using ELISA to measure Ig-specific antibody titers in plasma, as detailed in Section 3.8.

### 3.10. Antigen Uptake by Skin DCs: Flow Cytometric Analysis

OVA antigenic protein was labeled with a Cy5 kit (Cy5-OVA) using Cy5-monoreactive dye according to the manufacturer’s instructions. The tested sample containing FITC-OVA was identical to the sample containing Cy5-OVA except that Cy5-OVA was replaced with FITC-OVA. Antigen uptake by skin DCs, Langerhans cells (LCs), and lymph nodes was investigated by flow cytometry as previously described with slight modifications [14]. The monoclonal cellular assay experiment using flow cytometric methodology was examined using a multichannel flow cytometer, as briefly described in Appendix A.

### 3.11. Cytotoxic Immune Cell Counts in the TME

Mice were euthanized 7 days after the third dose of the vaccine, and the tumors were harvested and bisected. The desired tumor sections were washed in PBS and sectioned using a microtome, as described in the Appendix A. Cytotoxic (CD8+) T-cells in tumor tissues were quantified by staining nuclei with the fluorescent dye DAPI [43]. The number of CD8+ T-cells in in the TME was visualized and quantified using a fluorescence microscopic analyzer (BZ-900; Keyence, Osaka, Japan) and ImageJ software [4,44]. The TME observation is briefly described in Appendix A.

### 3.12. Statistical Analysis

Prism 6.0 software (Graph Pad Software, Inc., La Jolla, CA, USA) was used for all statistical analyses. For multiple comparisons, statistical significance was determined using one-way analysis of variance and Tukey’s post hoc test. Significance was indicated by *p* < 0.05.

## 4. Conclusions

Surfactants have a significant influence on LBIL-mediated pharmaceutical formulations, regulating formulation stability, toxicity, drug-loading capacity, and drug delivery. In this study, nonionic surfactants outperformed ionic surfactants. The combination of LBIL with a nanovaccine containing a nonionic-surfactant-associated protein and an immunomodulator significantly increased transdermal drug delivery by facilitating lipid arrangement changes in the SC, which increased prophylactic and therapeutic effects in the face of tumor challenge. The nanovaccine also significantly influenced antitumor immunity by stimulating the antigen uptake by dDCs and LCs, which increased the abundance of dermal CD207+, CD103+, and CD8+ T cells in the TME. These cytokines effectively destroy cancer cells and suppress tumor-associated immune-responsive diseases without inducing any toxic effects in cell models or drug-metabolizing organs. Notably, the nanovaccine has enormous clinical translation potential as a highly immunoresponsive TDDS enhancer that is completely biocompatible.

## Figures and Tables

**Figure 1 molecules-28-02969-f001:**
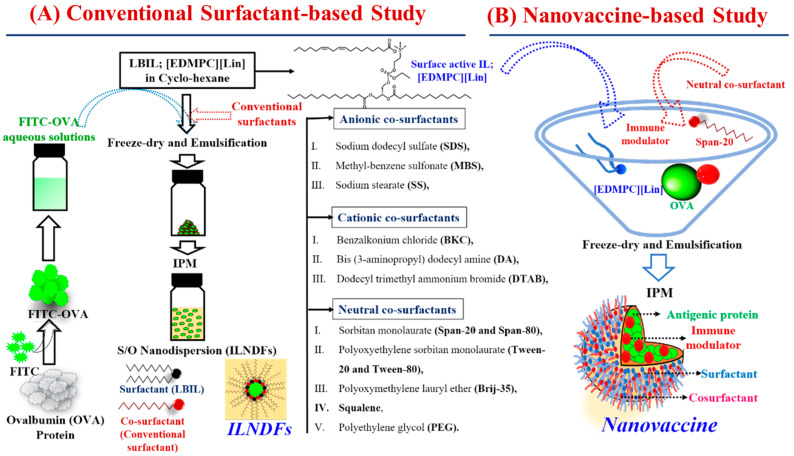
Schematically represented synthesis pathways of (**A**) ILNDFs and (**B**) IL-associated protein- and immunomodulator-containing nanoparticles for vaccination (nanovaccine).

**Figure 2 molecules-28-02969-f002:**
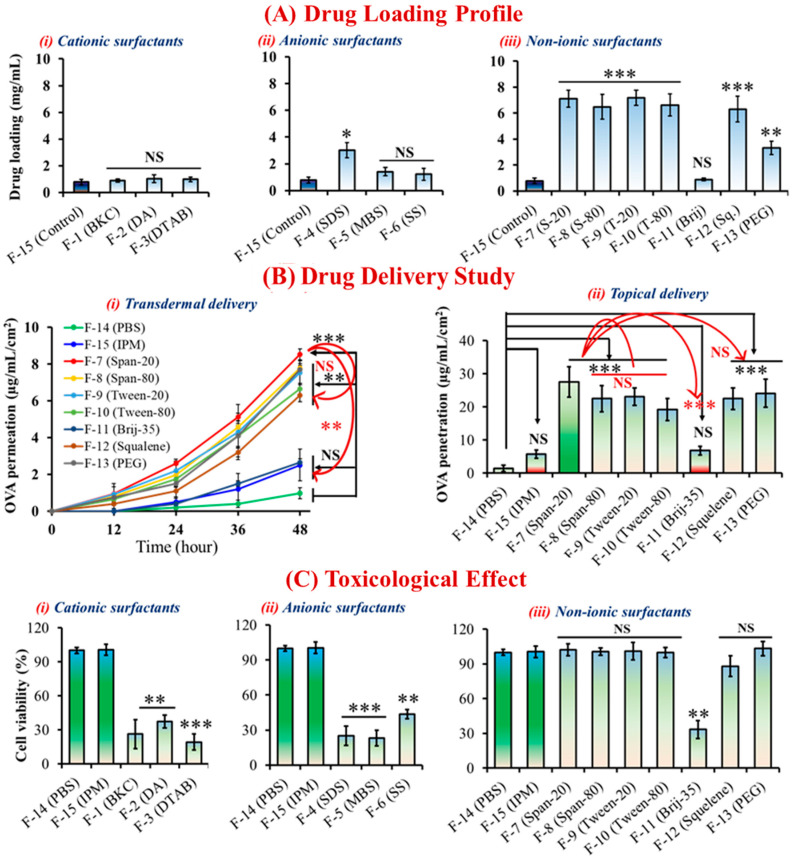
(**A**) Drug encapsulation capabilities of ILNDFs. (**B**) Transdermal and topical drug delivery of ILNDFs. (**C**) Cellular toxicity of ILNDFs. Data are presented as the mean ± SD (n = 3) * *p* < 0.1, ** *p* < 0.01, *** *p* < 0.001, and NS = not significant.

**Figure 3 molecules-28-02969-f003:**
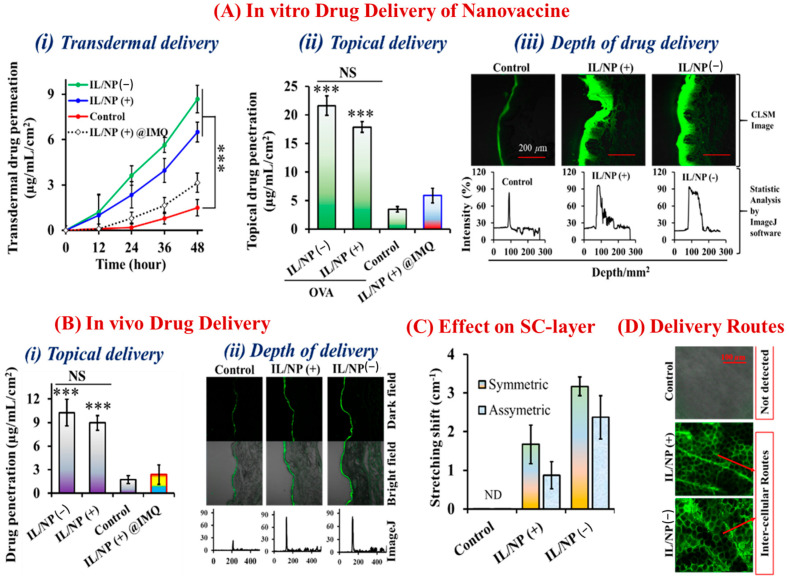
(**A**) In vitro (i) transdermal delivery, (ii) topical delivery, and (iii) depth of drug delivery into YMP skin via the nanovaccine. (**B**) In vivo topical delivery of FITC-OVA and IMQ via the nanovaccine in C57BL/6N mice. (**C**) Effect of the nanovaccine on the SC. (**D**) Intercellular route of FITC-OVA delivery. IL/NP(+) is the formulation containing IMQ, whereas IL/NP(−) is the formulation lacking IMQ. Data are presented as the mean ± SD (n = 3). *** *p* < 0.001, and IL/NP(+) @IMQ is a nanovaccine formulation without cosurfactant (S-20). NS = not significant, and ND = not detected.

**Figure 4 molecules-28-02969-f004:**
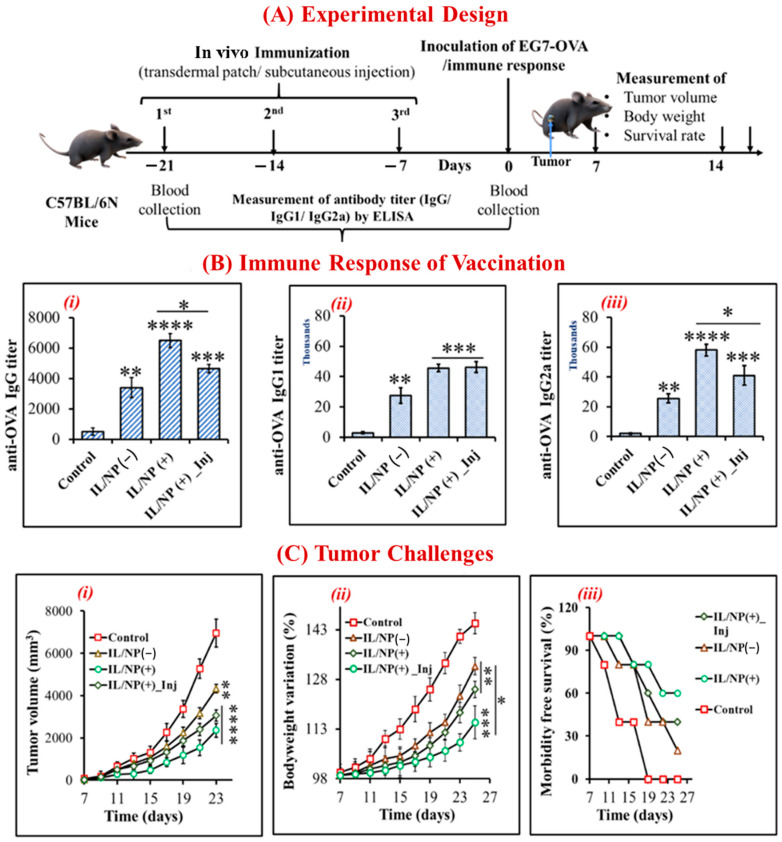
Prophylactic effect of the nanovaccine. (**A**) The experimental approach for vaccination in C57BL/6N mice. (**B**) OVA-specific IgG immune response to nanovaccination: (i) total IgG, (ii) IgG1, and (iii) IgG2a. (**C**) Antitumor effect of the nanovaccine on (i) tumor growth and development, (ii) body weight variation, and (iii) survival. Data are presented as the mean ± SD (n = 5). * *p* < 0.1, ** *p* < 0.01, *** *p* < 0.001, and **** *p* < 0.0001.

**Figure 5 molecules-28-02969-f005:**
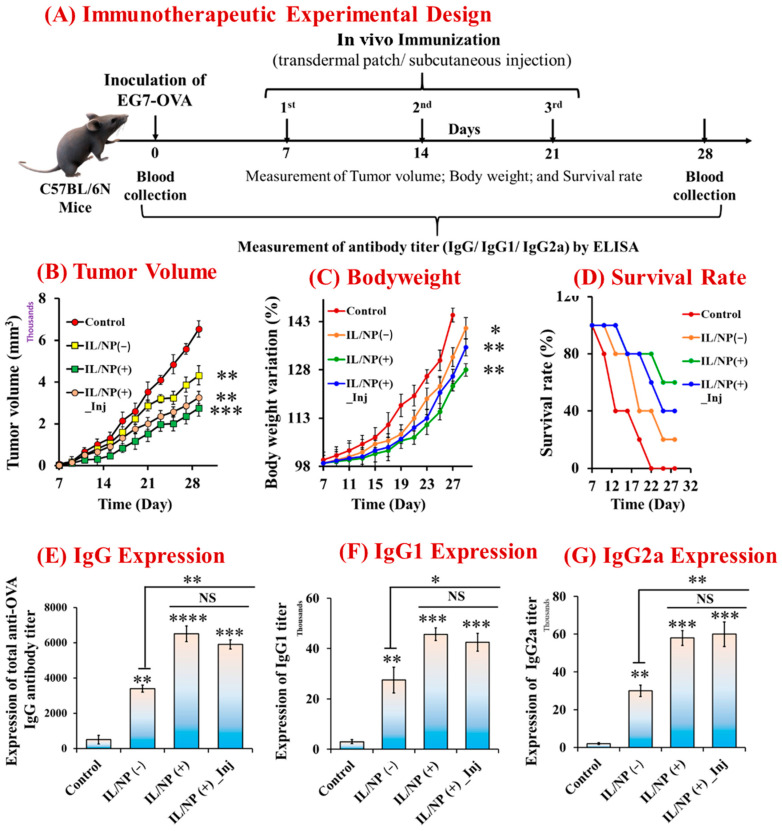
Antitumor effects of the protein- and immunomodulator-containing IL-associated nanovaccine. (**A**) Schematic illustration of the therapeutic experiment. (**B**) Effects on tumor growth and development. (**C**) Effect on body weight. (**D**) Effect on the survival rate. Effects of vaccination on OVA-specific antibody titers in bloodserum, including (**E**) total IgG, (**F**) IgG1, and (**G**) IgG2a. Data are presented as the ± SD (n = 5). * *p* < 0.1, ** *p* < 0.01, *** *p* < 0.001, **** *p* < 0.0001, and NS = not significant. Inj = administered via subcutaneous injection.

**Figure 6 molecules-28-02969-f006:**
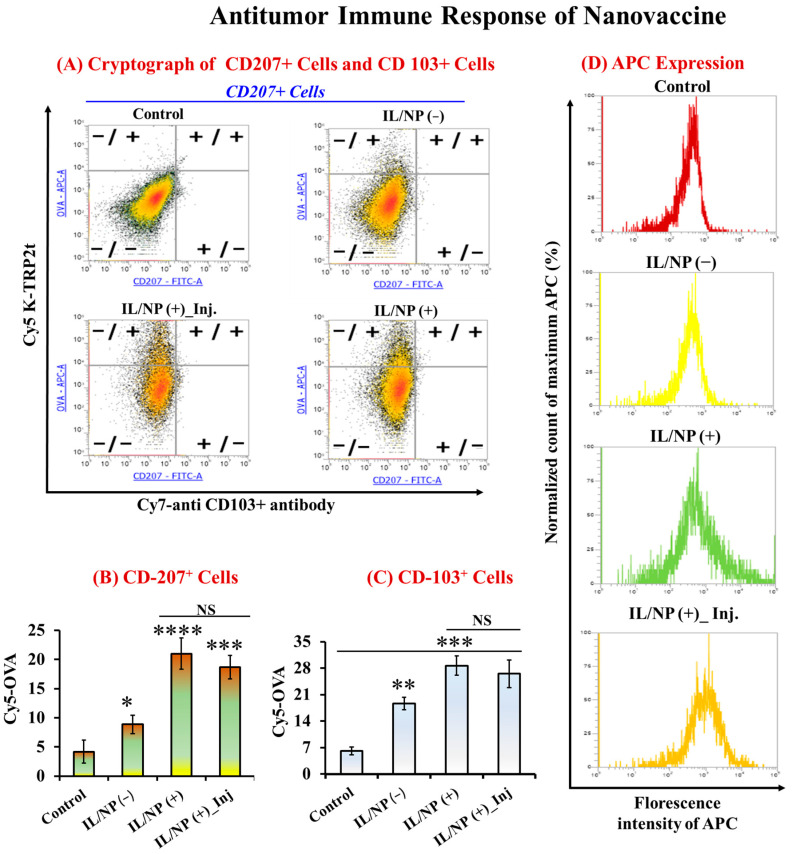
Antitumor immune response to the nanovaccine was investigated with flow cytometry using multichannel beam (**A**) cytograms, (**B**) Expression of the CD-207+ cells, (**C**) Expression of the CD-103+ cells, and (**D**) florescence intensity of APCs in skin tissues. Data are presented as the mean ± SD (n = 3). * *p* < 0.1, ** *p* < 0.01, *** *p* < 0.001, **** *p* < 0.0001, and NS = not significant. Inj = administered via subcutaneous injection.

**Figure 7 molecules-28-02969-f007:**
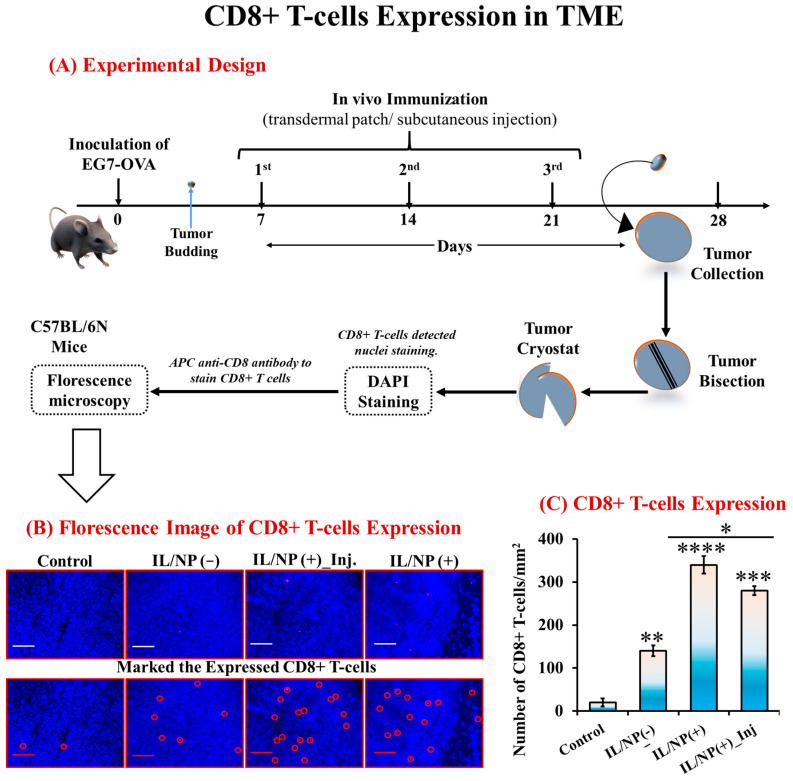
Number of CD8+ T cells in the TME following nanovaccination: (**A**) experimental design, (**B**) microscopic image, and (**C**) statistical analysis of CD8+ T-cell counts. Data are presented as the mean ± SD (n = 3). * *p* < 0.1, ** *p* < 0.01, *** *p* < 0.001, **** *p* < 0.0001, and Inj = administered via subcutaneous injection.

**Figure 8 molecules-28-02969-f008:**
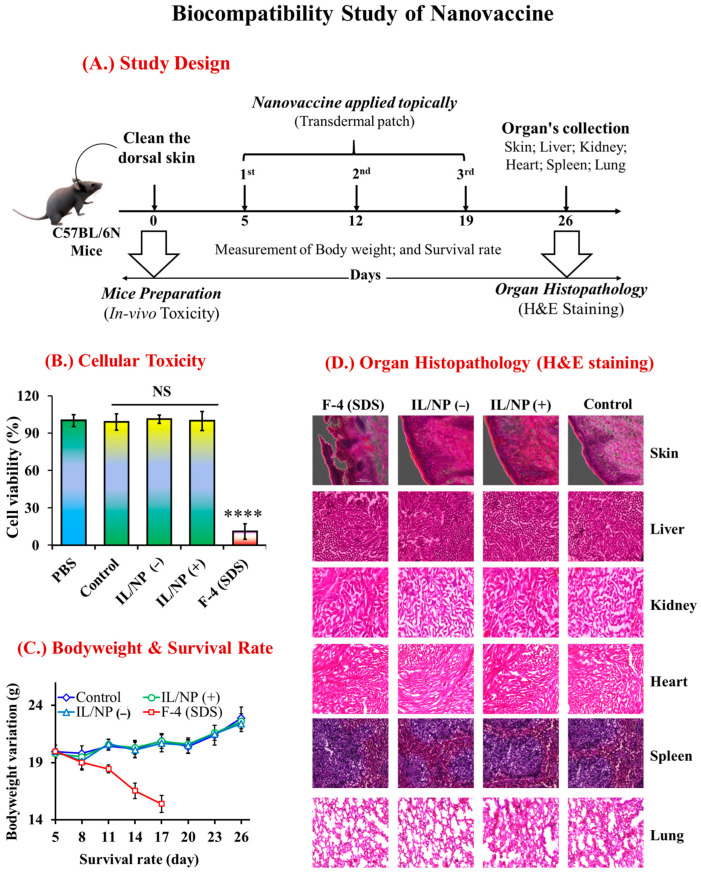
Biocompatibility study of the nanovaccine. (**A**) A schematic representation of the in vivo biocompatibility study. (**B**) A skin irritation test using the LabCyte EPI-MODEL cell line was performed with the MTT assay. (**C**) Body weight variation and survival rates of mice during the in vivo biocompatibility study. (**D**) Histopathological observation of the skin, liver, kidneys, heart, spleen, and lungs was performed by H&E staining. Data are presented as the mean ± SD (n = 3). **** *p* < 0.0001, and NS = not significant.

**Table 1 molecules-28-02969-t001:** Particle size and PDI of IL-containing conventional surfactant-based protein- and/or immunomodulator-containing formulations.

Formulations		Composition			Drug (mg/mL)		Particle Size (nm)	PDI
Name	Symbol *	Surfactant (2.5 wt.%)	Co-Surfactant (2.5 wt.%)	Base-Liquids (1 mL)	OVA (mg)	IMQ (mg)		
Cationic cosurfactants-based formulations								
IL-BKC-NDF	F-1	[EDMPC][Lin]	BKC	IPM	2	-	396	0.398
IL-DA-NDF	F-2	[EDMPC][Lin]	DA	IPM	2	-	841	0.562
IL-DTAB-NDF	F-3	[EDMPC][Lin]	DTAB	IPM	2	-	663	0.466
Anionic cosurfactants-based formulations								
IL-SDS-NDF	F-4	[EDMPC][Lin]	SDS	IPM	2	-	543	0.335
IL-MBS-NDF	F-5	[EDMPC][Lin]	MBS	IPM	2	-	712	0.621
IL-SS-NDF	F-6	[EDMPC][Lin]	SS	IPM	2	-	476	0.425
Nonionic or Neutral cosurfactants-based formulations								
IL-S-20-NDF	F-7	[EDMPC][Lin]	Span-20	IPM	2	-	260	0.254
IL-S-80-NDF	F-8	[EDMPC][Lin]	Span-80	IPM	2	-	429	0.301
IL-T-20-NDF	F-9	[EDMPC][Lin]	Tween-20	IPM	2	-	174	0.263
IL-T-80-NDF	F-10	[EDMPC][Lin]	Tween-80	IPM	2	-	216	0.212
IL-Brij-NDF	F-11	[EDMPC][Lin]	Brij-35	IPM	2	-	531	0.462
IL-Sq-NDF	F-12	[EDMPC][Lin]	Squalene	IPM	2	-	458	0.312
IL-PEG-NDF	F-13	[EDMPC][Lin]	PEG	IPM	2	-	824	0.391
Control group								
Aqueous NDF	F-14	-	-	PBS	2	-	-	-
Oil NDF	F-15	-	-	IPM	2	-	-	0.325
Formulations for nanovaccine study								
Control group	Control	-	-	PBS	2	1	-	-
IL/OVA+IMQ	IL/NP (+)_Inj.	-	-	PBS	2	1	-	-
IL/OVA+IMQ	IL/NP (+)_TP	[EDMPC][Lin]	Span-20	IPM	2	1	338	0.266
IL/OVA	IL/NP (−)	[EDMPC][Lin]	Span-20	IPM	2	-	269	0.309

* We used the symbol of the formulations in the relevant figures and tables. Inj. = administered via subcutaneous injection, TP = administered via a transdermal patch, (+) = formulation with IMQ, and (−) = formulation without IMQ.

**Table 2 molecules-28-02969-t002:** In vitro skin permeation kinetics of LCNDFs and the nanovaccine calculated using lag time methodology.

Formulation Symbol	Transdermal Flux	Lag Time	Permeation Coefficient	Diffusion Coefficient	Skin Partition Coefficient
	J (µg/cm^2^/h)	*t_L_* (h)	*K_p_* (cm/h)	*D* (×10^−3^ cm^2^/h)	*K_skin_*
Cationic cosurfactants-based formulations					
F-1 (BKC)	1.3 ± 0.2 ^a^	4.2 ± 0.5 ^NS^	0.03 ± 0.01	0.68 ± 0.02	0.08 ± 0.05
F-2 (DA)	2.1 ± 0.3 ^a^	3.7 ± 0.4 ^NS^	0.09 ± 0.05	0.72 ± 0.03	0.19 ± 0.07
F-3 (DTAB)	1.8 ± 0.5 ^a^	3.9 ± 0.6 ^NS^	0.06 ± 0.03	0.71 ± 0.02	0.15 ± 0.04
Anionic-cosurfactants-based formulations					
F-4 (SDS)	3.9 ± 0.8 ^b^	3.3 ± 0.6 ^a^	0.12 ± 0.04	0.63 ± 0.03	0.36 ± 0.02
F-5 (MBS)	1.4 ± 0.1 ^a^	3.8 ± 0.7 ^NS^	0.07 ± 0.03	0.06 ± 0.02	0.08 ± 0.02
F-6 (SS)	2.8 ± 0.4 ^a^	3.1 ± 0.3 ^a^	0.09 ± 0.02	0.46 ± 0.01	0.21 ± 0.05
Nonionic or neutral-cosurfactants-based formulations					
F-7 (Span-20)	6.6 ± 0.7 ^d^	2.8 ± 0.4 ^c^	0.15 ± 0.07	1.12 ± 0.09	0.68 ± 0.12
F-8 (Span-80)	5.1 ± 0.8 ^b^	2.8 ± 0.6 ^c^	0.14 ± 0.08	1.12 ± 0.11	0.52 ± 0.09
F-9 (Tween-20)	6.1 ± 0.5 ^c^	2.9 ± 0.5 ^b^	0.14 ± 0.05	1.11 ± 0.07	0.61 ± 0.13
F-10 (Tween-80)	5.4 ± 0.8 ^c^	2.9 ± 0.3 ^b^	0.12 ± 0.05	1.08 ± 0.05	0.49 ± 0.16
F-11 (Brij-35)	1.2 ± 0.3 ^a^	4.4 ± 0.8 ^NS^	0.04 ± 0.07	0.10 ± 0.02	0.07 ± 0.03
F-12 (Squalene)	4.8 ± 0.9 ^b^	3.2 ± 0.6 ^b^	0.11 ± 0.04	0.70 ± 0.05	0.39 ± 0.15
F-13 (PEG)	5.6 ± 0.7 ^c^	2.9 ± 0.5 ^b^	0.13 ± 0.09	0.98 ± 0.08	0.62 ± 0.22
Control group or without surfactant/cosurfactants-based formulations					
F-14 (PBS)	0.3 ± 0.1	4.5 ± 0.5	0.01 ± 0.01	0.04 ± 0.01	0.05 ± 0.02
Formulations for nanovaccine study					
Control	0.3 ± 0.1	4.5 ± 0.5	0.01 ± 0.01	0.02 ± 0.01	0.06 ± 0.01
IL/NP (+)	6.0 ± 0.6 ^c^	3.1 ± 0.3 ^a^	0.12 ± 0.05	1.10 ± 0.1	0.61 ± 0.09
IL/NP (-)	6.6 ± 0.7 ^d^	2.8 ± 0.4 ^c^	0.15 ± 0.07	1.12 ± 0.09	0.68 ± 0.12

^a^ *p* < 0.1, ^b^ *p* < 0.01, ^c^ *p* < 0.001, ^d^ *p* < 0.0001, and ^NS^ = not significant.

## Data Availability

All data generated or analyzed during this study are included in this article.

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
