# Peer review of "Modification with Conventional Surfactants to Improve a Lipid-Based Ionic-Liquid-Associated Transcutaneous Anticancer Vaccine"

_molecules, 2023, doi:10.3390/molecules28072969_

Round 1

Reviewer 1 Report

This paper deals with a transdermal antigen delivery technique using the ionic liquid mediated nano-drug carriers. The authors propose the interesting anticancer vaccine.

I have some comments that will improve the manuscript.

1.In Section 3.2, the authors mention that the ionic cosurfactant-based formulations were physical instability at room temperature. Please explain in detail on the reason for this.

2. In Section 3.3, the authors state the relationship between the hydrophilic-lipophilic balance of a surfactant and transdermal delivery. The HLB value of each surfactant should be clearly stated in this paper.

3. What is IL/NP(+)@IMQ described in Fig. 2?

4. The CLSM image in Fig. 3D seems to be the same as the one used in Ref. 4. Please check it carefully and revise it.

5. Figure 6A-i, iv needs to be revised to a clearer graph.

Author Response

Comments and suggestions for Author

This paper deals with a transdermal antigen delivery technique using the ionic liquid mediated nano-drug carriers. The authors propose the interesting anticancer vaccine.

I have some comments that will improve the manuscript.

Response: We thank the reviewer for the positive remark and constructive comments which have played an instrumental role in improving our manuscript. Based on your comments we have revised the manuscript as follows:

Comment-1: In section 3.2, the authors mentioned that the ionic cosurfactant-based formulations were physically instability at room temperature, please explain in detail on the reason for this.

Response: We value your suggestion. Based on this comment, we discuss potential causes for the stability and instability of ionic and non-ionic surfactants-based formulations. We also discuss the benefits and drawbacks of ionic and non-ionic surfactants in nanoparticles based liquid formulations which demonstrated either formulation stability and instability. We explain the reasons in results and discussion part in the section of 3.2. Morphological Behaviors and Cytotoxicity of ILNDFs. The changing portion was highlighted in red

Revised: page 8- 1st paragraph.

apart from Brij-35– and PEG-based formulations, were stable (Figure S1) [33]. The hydrophobic and hydrophilic structural groups of nonionic surfactants aid in correctly dispersing drug molecules in solution as emulsifiers or foaming agents, whereas ionic surfactants work in the reverse way, which leads formulation instability [34]. Non-ionic cosurfactants improved the phase stability with the IL and drug by forming covalent or weak van der Walls interactions in the oil phase, resulting in increased bioactivity and physiochemical stability for non-ionic ILNDFs [35]. Ionic surfactant can transfer its cationic or anionic charge to the surface of the nano-particles in liquid formulation, which leads the nanoparticles to deteriorate and precipitate due to electrostatic forces, as a results ionic surfactant-based formulation become unstable [36]. Moreover, ionic surfactants can be harsh and irritating to skin due to their interaction with the skin's and ions, whereas nonionic surfactants can reduce harshness, so they're skin-friendly [34]. Only non-ionic cosurfactant-based formulations were able to sustain particle size stability for 3 months (Figure S2-iii). The ionic properties of both cationic and anionic surfactants triggered precipitation, which led to instability. The interface attraction of surfactant and cosurfactant molecules acted as steric or electrostatic barriers to droplet particle agglomeration, consequently boosting formulation stability [37].

Comment-2: In section 3.3, the authors state the relationship between the hydrophilic-lipophilic balance of a surfactant and transdermal delivery. The HLB value of each surfactant should be clearly stated in this paper.

Response: We appreciate the reviewer’s insightful comments. We considerably improved our results and discussions sector (3.3), while attempting to establish a correlation between the HBL values of various surfactants (especially non-ionic cosurfactants) in drug formulations.  However, it is currently challenging to adequately describe how LBILs may affect pharmaceutical formulation, because the HBL values of LBILs have not yet been studied. To fully understand the mechanistic effect of LBILs in oil-based drug delivery systems, more research is required. The changing portion was highlighted in red.

Revised: Section 3.3, Page 9; Effect of ILNDFs on Drug Encapsulation and Skin Permeation; page 9- first paragraph.

Non-ionic surfactants have some advantages over ionic surfactants in that they can achieve a wide range of hydrophilic–lipophilic balances (HLB) by changing their molecular structure, in addition to their hydrophilic moiety, high degree of size customization, and lower critical micelle concentration, which is favorable for emulsification in the oil phase [39]. The HLB value represents the surfactant's propensity to incorporate in water or oil and form an emulsion.  Surfactants with low HLB values tend to be more soluble in oil, whereas those with high HLB numbers tend to be more soluble in water [40,41]. Those properties of non-ionic surfactants allowed them to load higher amounts of drug than ionic surfactants (Table S1). Compared with surfactant-free formulations, all non-ionic surfactants significantly increased (p < 0.001) the drug-loading capacity, whereas ionic surfactants had no such effect. Nonionic surfactants have varying HLB values (e.g., S-20 = 8.6, S-80 = 4.3, T-20=16.72, T-80 = 15, Brij-35 = 16.9, Squalene = 8.4, PEG = 20), and a reasonable range for emulsification is between 3.6 and 8.0 [36,40,41]. Owing to S-20's in a good range HBL values of 8.6 for emulsification formulations, it may predict that S-20's most potently increased drug-loading capacity over Tween and others (Figure 2A) [41]. However, it is currently challenging to adequately describe how LBILs may affect pharmaceutical formulation, because the HBL values of LBILs have not yet been studied. To fully understand the mechanistic effect of LBILs in oil-based drug delivery systems, more research is required. Surfactants have the ability to alter the permeability of cellular or biological membranes by concentrating at phase interfaces, resulting in lower membranous interfacial tension with an increasing duration of skin contact, which influences transdermal drug delivery [37,39].

Comment-3: What is IL/NP(+) @IMQ described in Fig 2?

Response: We appreciate this suggestion.! In fact, this term is used in Figure 3. We have considered it as figure 3 and clarified about the term of “IL/NP(+) @IMQ” in figure 3 and mentioned it in the text of Figure 3. We also clarified it in the results and discussion sector of 3.4. Morphological and Transdermal Delivery of the Nanovaccine.

Revised: In the text of Figure 3, page 11; IL/NP(+) @IMQ is a nanovaccine formulation without cosurfactant (S-20).

Revised: In discussion section 3.4, page 10-11; (Only IL-surfactant (LBIL)-based formulation IL/NP (+) @IMQ had less effective transdermal drug delivery and formulation stability than the formulation coated with surfactant (LBIL) and cosurfactant (S-20); IL/NP(+) (Figure 3A-i,ii and Figure 3B-i). The findings demonstrated that S-20 cosurfactant was employed in formulation to enhance the stability of the drug formulation as well as the drug release profile.)

Comment-4: The CLSM image in Fig. 3D seems to be the same as one used in Ref.4. Please check it carefully and revised it.

Response: We value your astute observation. Even while it seems like the CLSM image in Fig. 3D and the one in Reference 4 are the same, but both are not exactly same as. We employed the same fluorescent molecules (FITC) and CLSM technology in this investigation and Ref-4. When we tuned a Confocal laser scanning microscopy (CLSM) instrument using the same parameters for different samples, it might have caused a similar snap for organelles of a similar type. Regrettably, there were many similarities with one image in Ref-4 and this article.  We altered and redrew the image in Figure 3D in order to prevent confession.  During our studies, we snap several images for a sample, and we added another CLSM image from them in Figure 3D.

Comment-5: Figure 6A-I, iv needs to be revised to clear graph.

Response: In response to the reviewer's remarks, we implemented the required updates in figure 6. Furthermore, we separated it into two figures and constructed those figures with clearer images. Former figures 6A and 6B have been changed to figures 6 and 7, respectively. When we joined the new figures 6A and 6B, it resulted figure seems in very large sizes, therefore we divided these two parts into two independent figures. Following these modifications, the previously stated figure 7, now referred to as figure 8, and the entire manuscript are rearranged accordingly.

Revised:

Page 16, Figure 6. Antitumor immune response to the nanovaccine was investigated by flow cytometry using multichannel beam (i) cytograms, (ii) number of CD-207+ cells, (iii) number of CD-103+ cells, and (iv) florescence intensity of APCs in skin tissues. Data are presented as the mean ± SD (n = 3). *p < 0.1, **p < 0.01, ***p < 0.001, ****p < 0.0001, and NS = not significant. Inj = administered via subcutaneous injection.

Page 17, Figure 7. Number of CD8+ T-cells in the TME following nanovaccination: (i) experimental design, (ii) microscopic image, and (iii) statistical analysis of CD8+ T-cell counts. Data are presented as the mean ± SD (n = 3). *p < 0.1, **p < 0.01, ***p < 0.001, ****p < 0.0001, and NS = not significant. Inj = administered via subcutaneous injection.

Page 19, Figure 8. Biocompatibility study of the nanovaccine. (A) A schematic representation of the in vivo biocompatibility study. (B) A skin irritation test using the LabCyte EPI-MODEL cell line was performed by the MTT assay. (C) Body weight variation and survival rates of mice during the in vivo biocompatibility study. (D) Histopathological observation of the skin, liver, kidneys, heart, spleen, and lungs was performed by H&E staining. Data are presented as the mean ± SD (n = 3). *p < 0.1, **p < 0.01, ***p < 0.001, ****p < 0.0001, and NS = not significant.

Reviewer 2 Report

The article screen different surfactants for ionic liquid based nanoparticle formulation. The authors have done through analysis of their formulations by several in vitro and in vivo tests. The formulation screen does help in identifying the best surfactant for the vaccine formulation being developed. However, there are some minor issues that need to be addressed. 1. Authors need to give an introduction on what are ionic liquids and how usually they are formulated and used in medicine. 2. Authors need to improve sentence structures in the introduction. Also, it is advised to refrain from making statements that are not supported by the reference. Like “Transcutaneous vaccination is the most successful, affordable, and patient-friendly advanced immunization approach because of the presence of multiple immune-responsive cell types in skin.”.  Transcutaneous vaccination is not the most successful vaccination approach. The introduction and discussion section has many such examples and they need to be corrected. 3. The authors do not discuss about how less efficacious the cancer vaccine is. The lead formulation is just slowing down the tumor growth to such a less extent and the vaccine is not able to induce a response good enough to wipe out the tumor which is actually normal for an anticancer vaccine.

Except for these three main points, the research article is good at addressing important points in terms of design, method, results and discussion. 

Author Response

Comments and Suggestions for Authors

The article screen different surfactants for ionic liquid-based nanoparticle formulation. The authors have done through analysis of their formulations by several in vitro and in vivo tests. The formulation screen does help in identifying the best surfactant for the vaccine formulation being developed. However, there are some minor issues that need to be addressed.

Response: We appreciate the reviewer's helpful suggestions and complimentary remarks, which helped us refine our manuscript. We've made the following changes to the manuscript in response to your feedback: Red colour is used to identify the modified parts.

Comment- 1. Authors need to give an introduction on what are ionic liquids and how usually they are formulated and used in medicine.

Response: We acknowledge your suggestion. In response to your comments, we developed the section of our manuscript's introduction where we discussed about ionic liquids and how they are employed in medicinal formulations.

Revised:

1st para of Introduction section; Page 2: Ionic liquids (ILs), which are liquid salts with melting temperature below 100 °C, can be used as reagents, solvents, and anti-solvents in the synthesis and crystallization of active pharmaceutical ingredients (APIs), as solvents, co-solvents, and emulsifiers in drug formulations, as pharmaceuticals (API-ILs) aiming at liquid therapeutics, and in the development and/or improvement of drug-delivery-based systems. Recently, new developed lipid-based biocompatible ionic liquids (LBILs) actively employed in nano-drug formulations (ILNDFs) represent an emerging modality for overcoming these challenges and improving transdermal drug delivery systems (TDDSs) based on the lipophilic and wide solubility of LBILs.

Comment- 2. Authors need to improve sentence structures in the introduction. Also, it is advised to refrain from making statements that are not supported by the reference. Like “Transcutaneous vaccination is the most successful, affordable, and patient-friendly advanced immunization approach because of the presence of multiple immune-responsive cell types in skin.”.  Transcutaneous vaccination is not the most successful vaccination approach. The introduction and discussion section has many such examples and they need to be corrected.

Response: We value your suggestion. We attempted to improve the sentence structure in the introduction, although our initial manuscript was edited by professional native English speaker Joe Barber Jr., PhD, from Edanz (https://jp.edanz.com/ac). We responded to the reviewer's comments and made the necessary improvements.

Revised: First part of Abstract, Page1: Transcutaneous vaccination is one of the successful, affordable, and patient-friendly advanced immunization approaches because of the presence of multiple immune-responsive cell types in skin.

Comment- 3. The authors do not discuss about how less efficacious the cancer vaccine is. The lead formulation is just slowing down the tumor growth to such a less extent and the vaccine is not able to induce a response good enough to wipe out the tumor which is actually normal for an anticancer vaccine.

 Response: We appreciate the reviewer’s insightful comments. We considerably improved the introduction, results and discussions sectors. Red colour is used to identify the modified parts.

In our current study, we investigated the pharmacokinetics profile and the anticancer profile of the LBIL-associated nano vaccine to see how it affected tumor reduction. For anticancer profile, we observed that our developed nano-vaccine significantly slowing down the tumor growth and development compare to its aqueous formulation. We are unable to compare our developed nano-vaccine formulation with any commercial anticancer medication formulations due to the lack of OVA-specific anticancer drugs on the market. Our research into the therapeutic and preventative potential of nanovaccine revealed that it dramatically enhanced the body's auto immune response and raised the expression of antitumor immune cells (such as IgG, CD cells). Even our newly created nanovaccine demonstrated superior anti-cancer immune responses when administered transcutaneous as opposed to injection-based.

In our manuscript, we discussed those statements in detail in the flowing parts,

2.8. Vaccination and Immune Response against the Tumor:

2.9. Prophylactic and Therapeutic Effects of the Nanovaccine against Tumors

2.10. Antigen Uptake by Skin DCs: Flow Cytometric Analysis

2.11. Cytotoxic Immune Cell Counts in the TME

3.5. Prophylactic Effect of the Nanovaccine against Tumors

3.6. Therapeutic Immunization and Tumor-suppressive Effects of the Nanovaccine

3.7. Antitumor Immune Response Induced by the Nanovaccine

Revised: Page 14; 1st para. And Page 14; last para.

Except for these three main points, the research article is good at addressing important points in terms of design, method, results and discussion. 
